# DCGAN-based synthetic image generation of denim jeans defects

Muhammad Naeem[1], Qaisar Abbas[2], Haseeb Ahmad[1], Muhammad Salman Naeem[3], Mutlaq B. Aldajani[2], Hussain Dawood[4] and Muhammad Awais Hussain[5]

[1] Department of Computer Science, National Textile University, Faisalabad, Pakistan
[2] College of Computer and Information Sciences, Al-Imam Mohamed Ibn Saud Islamic University, Riyadh, Saudi Arabia
[3] School of Art and Design, National Textile University, Faisalabad, Pakistan
[4] School of Computing, Horizon University College, Ajman, United Arab Emirates
[5] Wenzhou Institute of Zhejiang University, Wenzhou, China

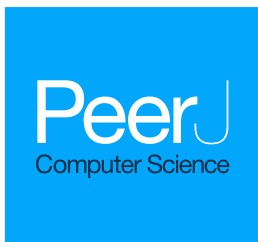

Corresponding author
Haseeb Ahmad,
haseeb_ad@ntu.edu.pk

## ABSTRACT

**Background:** Automated defect detection in denim jeans manufacturing is crucial for maintaining quality control efficiency. However, for automated defect detection of denim jeans, machine learning algorithms suffer due to limited data availability because manufacturing industry remains reluctant to share data due to privacy concerns. Moreover, remote manufacturing units make it more difficult to gather necessary defected images. Furthermore, trained personnel are required to capture standard images for training effective models. Traditional image augmentation approaches extend the datasets from seed images; however, there is a lack in image diversification and they do not expand data distribution and thus may lead to overfitting. Deep learning models, especially generative adversarial networks, have the potential to provide effective solutions for industrial problems, such as synthetic image generation for denim jeans defect detection.

**Methods:** This article proposes the use of a deep convolutional generative adversarial network (DCGAN) for generating diversified and realistic synthetic images of common denim jeans defects, including broken loops, broken stitches, skipped stitches and twisted legs. The DCGAN model was trained on an initial dataset of 3,930 defective images and subsequently augmented using techniques such as flipping, random zooming, and color space augmentation.

**Results:** The generated synthetic images were subjectively validated by domain experts, achieving an average accuracy of 81.5% Objective evaluation using the Fréchet inception distance metric also demonstrated the effectiveness of the proposed approach, with scores of 12.26, 6.75, 7.68 and 27.59 for broken loop, broken stitch, skipped stitch and twisted leg defects, respectively. This work not only contributes to addressing the challenge of data scarcity in defect detection but also paves the way for more accurate automated defect detection systems in denim jeans manufacturing.

# INTRODUCTION

The impact of artificial intelligence (AI) and data driven decision making on the manufacturing industry has far exceeded expectations. Research and development teams are actively working to advance the field of industrial intelligence. In 2010, Germany introduced the famous Industry 4.0 framework that has gained widespread adoption globally. This proliferation has immensely supported the inevitable shift of manufacturing industry to smart industry (*IBM, 2024*). This paradigm shift has urged the textiles industry to modernize its manufacturing processes (*Jiang et al., 2024*). Textile production is a large-scale complex industry entailing a series of intricate and systematic processes including spinning, weaving, dyeing, printing, finishing, and apparel manufacturing (*Tuna, 2018*). Apparel production consists of a chain of activities including fabric inspection, cutting, assembling of small parts, stitching of wearables, and finally the quality assurance (*Ren et al., 2022*).

Ensuring the quality assurance of stitched denim jeans, produced through the textile process is paramount in the apparel manufacturing industry. Though factors including the fastness properties of dyed fabric, application of finishing materials influence the quality of the finished product, however, the stitched apparel is mostly affected by the defects appear during the stitching and assembling process. These defects may lead to the quality degradation and therefore, result in wasted cost, time, and resources (*Liu & Zheng, 2021*). Moreover, manual defect detection is prone to imprecision and often requires more time for quality assurance. Such issues usually damage the fame of export industry and lead to lose the trust of customers (*Nadhif & Kusumawardhani, 2021*). Therefore, it is imperative to implement an effective and automated defect detection mechanism in the denim jeans stitching industry to mitigate the aforementioned issues.

Automated defect detection mechanisms offer a promising solution ensuring quality parameters in denim jeans production. Manual defect detection process currently practiced in large scale industries is a time-consuming inefficient process (*Toan, 2022*). In contrast, automated inspection methods can reduce the inspection costs, improve defect detection accuracy and increase overall productivity (*Kim et al., 2022*). Research over the past decade has primarily focused on developing defect detection algorithms, but most efforts are made for fabric defect detection rather than apparel. Automated defect detection models are hot topics in industry and scientists are more focused on solving industrial needs. Several fabric defect detection algorithms are proposed recently demonstrating exceptional performance in commercial manufacturing processes. These algorithms can be broadly categorized into two classes including traditional and learning-based algorithms (*Wu et al., 2021*). Traditional algorithms typically focus on feature construction using available knowledge including model-based, structural, spectral, and statistical models (*Dong et al., 2020*). In contrast, learning-based algorithms include conventional machine learning and deep learning algorithms (*Wu et al., 2021*). Machine learning is used for classifications and predictions using mathematical and statistical formulations that can be

further used for decision-making. Learning based algorithms also includes data-driven and self-supervised methods (*Wang & Liu, 2024*).

Learning-based algorithms have become increasingly important in research across various domains. Active and incremental learning techniques, for instance, have been applied to network anomaly detection (*Tian et al., 2023*). Similarly, deep learning algorithms, including classification and segmentation, are employed in the state-of-the-art for fabric detection with promising results. Single-stage and double-stage object detectors are also used in different studies (*Jing & Ren, 2021*). More precisely, single-stage object detection is simpler and faster, however, it may compromise accuracy (*Rahimunnisa, 2022*). Conversely, double stage process provides higher accuracy at a slower rate. These deep learning techniques are the core of quality control processes. However, training deep learning models requires extensive defected and non-defected images. Acquiring such thousand images from various industries over extended periods can be challenging due to factors such as brightness and resolution, capturing diversity, and disrupting the manufacturing process and industrial-academia gaps. Additionally, privacy concerns of manufacturers may hinder data collection (*ul-Huda et al., 2024*). The scarcity of defective and non-defective images data limits the effectiveness of defect detection and classification models, ultimately hindering automated defect detection in denim jeans manufacturing. Insufficient training data also hurdles the research and advancements in the field under discussion. To address these challenges, synthetic data generation using deep learning models may be explored.

Data augmentation techniques such as geometrical transformations, color transformations, histogram sliding, cropping, kernel filters, flipping and noise addition are employed to expand the datasets (*Awan, 2024*). Popular tools for data augmentation include Pytorch, Augmenter, Albumentation, Imgaug and OpenCV. While augmentation may yield satisfactory results, however, it may lead to model overfitting and may not enhance the data diversity or robustness. Synthetic data generation using techniques like generative adversarial network (GAN), introduced in 2014 (*Goodfellow et al., 2014*), offer potential solutions. GANs, a widely used combination of two deep neural networks (generator and discriminator), have spawned numerous variants such as Cycle GAN (*Wen et al., 2021*), CartoonGAN, Zoom GAN and discriminator-guided learning using vanillaGAN DGL-GAN.

This research proposes a deep convolutional generative adversarial network (DCGAN) for synthetically generating denim jeans defect image data. More precisely, the DCGAN model is trained to generate four types of denim jeans defects: broken loops, skipped stitches, twisted leg and broken stitches. By leveraging the proposed model, a large dataset of various denim jeans defects may be synthesized, which may then be used to train automatic defect detection and classification models, enhancing their accuracy and precision. This research aims to address several questions, including: Can a single DCGAN architecture effectively produce different types of defect images? Which augmentation techniques support effective data generation with DCGAN? How do parameters

adjustments impact learning quality and generation effectiveness? Which metrics may be used to evaluate DCGAN performance? What are the future directions for improving the DCGAN model for denim jeans defects generation. Specifically, this work contributes the following:

- To capturing a preliminary dataset of denim jeans defects including broken loops, broken stitches, skipped stitches and twisted leg.
- To develop an effective DCGAN model for synthetically generating detailed, diversified and quality images of denim jeans defects.

## Theoretical background

This section provides a concise overview of GAN architecture including the structure of generator and discriminator networks, their respective loss functions, and the underlying principles of GANs. Subsequently, it delves into common defects found in denim jeans. Additionally, the section explores the significant contributions of deep learning to automated defect detection and classification systems.

### Generative adversarial networks

GANs represent an unsupervised learning approach that focuses on the learning of underlying data representations rather than relying heavily on annotated training data. A GAN consists of two competing networks trained through backpropagation. GANs have applications in synthetic image generation, styling, semantic image editing, data augmentation and super resolution (*Creswell et al., 2017*). A typical GAN architecture involves a generator (G) and a discriminator (D). The generator acts as a 'forger' producing synthetic samples, while the discriminator functions as an 'expert' trying to distinguish between real and fake samples. The generator learns from the discriminator's feedback, while the discriminator is trained on both real and synthetic samples. Both generator and discriminator are convolutional networks. The generator maps from a latent space, represented as G: G(z) → R, where z is a sample from the latent space. The discriminator classifies samples as real or fake, providing a probability (0–1) of the sample being real. After optimal training, the discriminator's role is typically halted, and the generator continues to produce synthetic samples.

### Generator training

The generator network in GAN is responsible for generating synthetic samples that aim to deceive the discriminator. The generator seeks to minimize its loss function $J_G$ by maximizing the log probability that the discriminator perceives the generated samples as real. The generator's loss function is mathematically expressed as follows:

$$J_G = -1/m \sum_{(i=1)}^{m} \log D(G(z_i)), \tag{1}$$

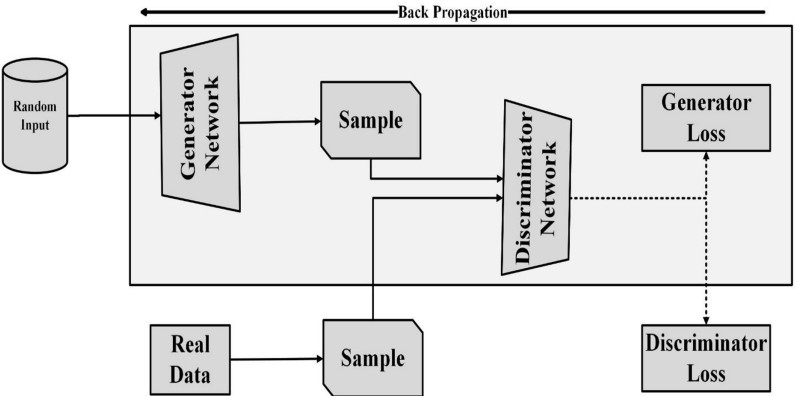

**Figure 1 Training of generator in GAN.**

where, the performance of the generator (in deceiving the discriminator) is measured in terms of $J_G$ by $\log D\,(G(z_i))$, the log probability of the discriminator correctly perceiving the generated samples. The generator aims to minimize this loss by producing samples that the discriminator misclassifies as real. Figure 1 illustrates the working and backpropagation process of the generator network.

## Discriminator training

The discriminator's success lies in its ability to accurately distinguish between real and synthetic samples. It aims to minimize the negatively signed log likelihood of correctly identifying both generated and actual samples. This loss function encourages the discriminator to classify synthetic and real data accurately. The discriminator's ability to recognize generated and actual samples is measured by JD, as mathematically demonstrated in Eq. (2):

$$J_D = -1/m \sum_{(i=1)}^{m} \log(D(x_i)) - 1/m \sum_{(i=1)}^{m} \log[\![(1 - D(G(z_i)))]\!], \qquad (2)$$

where $J_D$ is the loss function of the discriminator network in which $\log D(x_i)$ represents the log-likelihood of discriminators accurate identification (*Zhao et al., 2024a*). The expression $\log(1 - D(G(z_i)))$ denotes the log probability of discriminators wrong identification. Contrary to the generator, the discriminator wants to reduce the loss by accurately identification between generated and actual samples as shown in Fig. 2.

The generator (G) aims to generate samples that the discriminator cannot accurately classify as real or fake. Conversely, the discriminator seeks to accurately classify both real and generated samples. This adversarial relationship between the generator and discriminator can be viewed as a minmax game, resulting in a GAN. Through this adversarial process, the generator learns a probability distribution that effectively deceives the discriminator. Once the generator has mastered this, it can produce samples that the

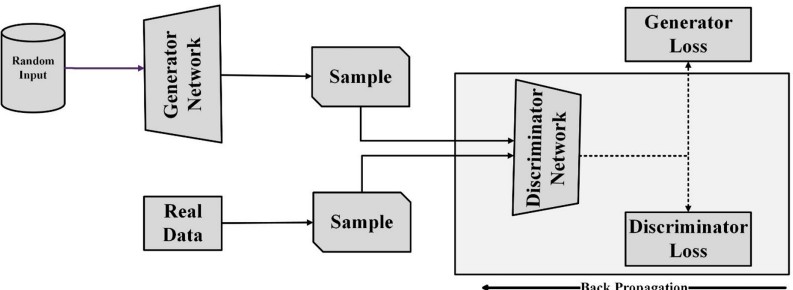

**Figure 2 Training of discriminator in GAN.**

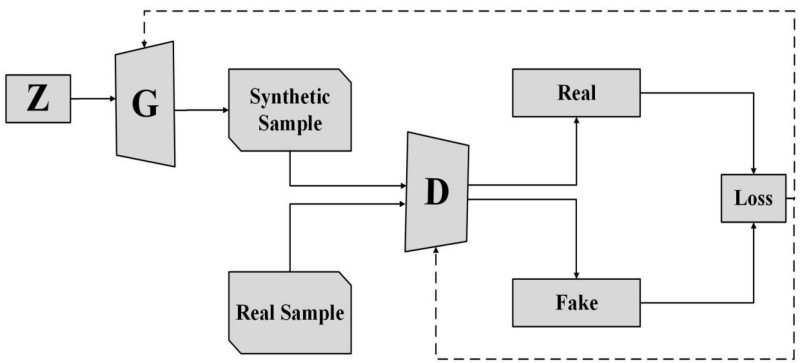

**Figure 3 GAN architecture.**

discriminator cannot distinguish as real or fake. This minmax game can be mathematically expressed as follows:

$$(min)_\top G(max)_\top DV(D,G) = E_{(x \sim p''_{dt''}(x))}[logD(x)] + E_{(z \sim p_z(z))}[log(1 - D(G(z)))], \qquad (3)$$

where x are the actual samples containing the true data distribution presented as $E_{(x \sim p''_{dt''}(x))}$. Random noise provided to the generator is denoted as term $p_z(z).D(x)$ is discriminator probability of accurate identification, whereas $D(G(z))$ is the probability that discriminator perceive the synthetic data as real.

*Radford, Metz & Chintala (2016)* proposed first DCGAN as an advancement over basic GAN architecture. Unlike traditional GANs, DCGANs employ deep convolutional neural networks (DCNNs) to extract features from sample images, resulting in faster training and improved performance. DCGAN architecture includes convolutional and transposed convolutional layers. DCNNs are used instead of multi-layer perceptrons, and batch normalization is used between convolutional layers for stability. Both generator and discriminator networks are trained concurrently. The generator network generates synthetic images from random input (noise) with the aim of deceiving the discriminator. The discriminator network is responsible for accurately classifying generated images as real or fake. To fool the discriminator, the generator model learns the probability distribution of the real-world dataset. After training, the generator network effectively learns this distribution and starts generating synthetic data that the discriminator perceives

as real. Figure 3 illustrates the GAN architecture, showcasing how the generator generates synthetic data from random input.

*Radford, Metz & Chintala (2016)* employed stride convolutions in both the generator and discriminator for spatial upsampling and replaced fully connected layers with global average pooling. A noise vector (Z) was provided as input to the first fully connected layer of the GAN, producing a 4-D tensor output. Batch normalization was used to standardize input units, followed by Tanh activation in the output layers and ReLU activation in other layers of the generator. LeakyReLU was used in all discriminator layers. The model was trained and tested on the LSUN bedroom dataset.

Synthetic data generation is not limited to images but has also been applied to other data types, such as textual, structural, and hierarchical data. *Park et al. (2018)* used Table-GAN to generate fake data for addressing privacy concerns in organizations. The Table-GAN architecture includes a classifier to control the statistical properties of the original data, ensuring that the generated dataset maintains similar characteristics like mean and standard deviation. Datasets from various sources, including LACity (Los Angeles City Government Employees), health department, and airline data, were used to evaluate the model's performance. The results were compared to other generation techniques like perturbation, anonymization, and other generation methods. GANs have also found applications in the medical field. For example, GAN-cAED, a variant of GAN (*Cheema et al., 2021*), has been developed to assist doctors in preventing accidental cuts to major blood vessels. GANs are also being used for defect detection purposes. A multiscale inpainting GAN has been used to detect the defects in industrial surfaces. The multimodal feature fusion-based generative adversarial network (M-GAN) has been used to encompasses the dataset challenges like class imbalance problem, multimodel feature fusion and entanglement between discriminator and generator nets in a GAN (*Zhao et al., 2024b*). Similarly, the balance structure conditional (*Wan, Zhou & Wang, 2024*) GAN has been used to overcome the class imbalance problem. GANs have also contributed to generating motion videos of human beings using a few images using 2D and 3D pose estimations (*Kumar & Singh, 2024*). A lightweight deep fusion attention feature model has been used for the text to flower image creation. The model used a target aware discriminator in GAN capable of training on rich feature mapping, it has enhanced the ability of showing details and visual uniformity in text to flower images (*Yang et al., 2024*). Alongside the GAN working with a random noise, pix to pix (*An, Wu & Zhang, 2024*) GANs like occlusion aware segmentation using RCF have used to compare an image sample to generate another similar image sample. Generative techniques have also been applied to construction-related applications, such as image quality improvement, building design and architecture creation, and generation of concrete crack images.

## Defects of denim jeans

Quality assurance and product enhancement are essential for ensuring that products meet specified requirements. Inspection, a crucial quality control method, is used to verify that products conform to expected quality standards. The goal of inspection is to identify defective products as early as possible. Denim jeans production, including the stitching

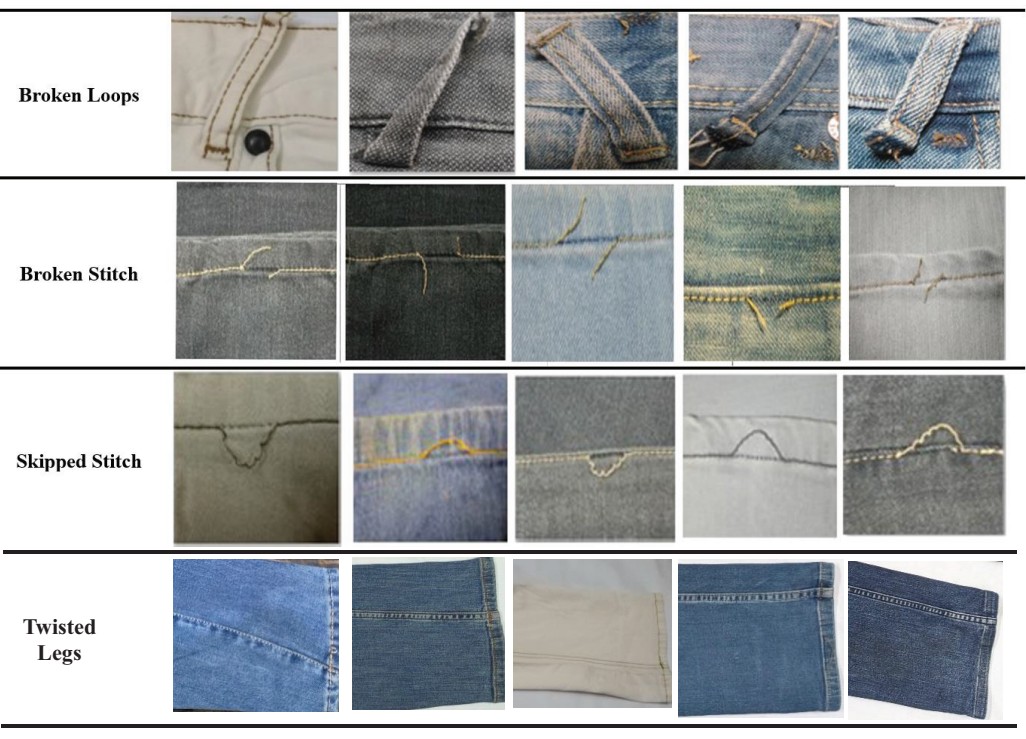

**Figure 4  Original defect images.**

process, is a significant sector of the textile industry. However, production processes may sometimes yield unexpected results due to factors like improper skills, faulty machines, or inappropriate machine adjustments. These defects can negatively impact quality and increase production costs. To maintain quality standards, it is crucial to detect defects before shipment or export. Common defects encountered in denim jeans manufacturing include broken stitches, broken stitch abrasion, unrevealing seams, drop stitches, ropy hems, and twisted legs. The specific defects targeted for synthetic generation in this study are visually depicted in Fig. 4.

A study conducted by the Faculty of Business and Economics at Diponegoro University, Semarang, involved a survey of the apparel industry (*Taner Ersöz et al., 2021*). The number of detected defects was represented using histogram. Approximately 36,000 units were tested, revealing around 3,500 defects, as shown in Fig. 5. These results highlight the significant time and effort invested in the monitoring process. Furthermore, the study indicates a defect rate of approximately 10%. An automated system can not only save time and costs but also ensure compliance with quality standards.

The presence of defects in denim jeans can significantly impact the quality standards of production organizations. Manually detecting these defects through visual inspection is a challenging, time-consuming, and costly task. Similar to the automation of other processes and mechanisms, automating denim jeans defect detection requires deep learning-based models for effective detection and classification.

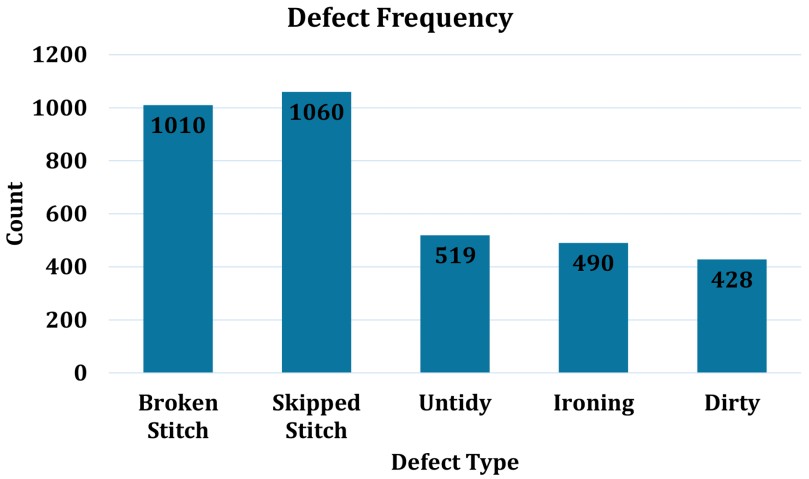

**Figure 5** Frequency of defects in stitched apparels.

## Deep learning contributions towards defect detection

Numerous studies have focused on fabric defect detection, a prominent research area in the past two decades. Several algorithmic models have been proposed for detecting threads and stitches, following fabric inspection. In 2022, a defect detection algorithm was introduced for the garment manufacturing industry (*Kim et al., 2022*). This study employed deep learning methods, specifically convolutional neural networks (CNNs). CNNs demonstrated satisfactory performance by utilizing feature maps from the convolutional layers. These layers incorporated a pre-trained visual geometry group (VGG)-16 deep neural network. VGG-16 was used to extract feature maps from the initial layers of the CNN. The performance was evaluated on a dataset of images containing normal sewing, artificial defects, and rotated images. The proposed method achieved high accuracy in detecting broken stitches.

In 2021, Yavuz Kahraman employed the visual geometry group (VGG)-19 concepts for defect classification. A similar study at Karabük University, Turkey, focused on detecting fabric and production process defects in textile apparel industries using data mining techniques like Random Forest, naive Bayes, Decision Tree, and gradient descent. A study (*Voronin et al., 2021*) proposed an automated defect detection approach through visual inspection of apparel using deep learning. This approach involved a two-phased process: local and global domain-oriented image contrast enhancement using alpha rooting, followed by a modern artificial neural network (ANN) architecture for defect detection. This solution demonstrated superior accuracy compared to traditional algorithms. The TILDA data repository was used to evaluate the model's effectiveness.

A study conducted at North Carolina State University, Raleigh, NC, USA, in 2021 aimed to detect defects in fabric printing (*Nilsson & Lindstam, 2012*). The study utilized a deep convolutional neural network by integrating the VGG network with DenseNet, Inception, and Exception deep networks. The research process involved database development, dataset splitting, image amplification, and hyperparameter selection. The developed model

was then executed on a database of fabric images for classification. The model successfully identified defective and non-directive printed fabrics with misprints and color spots (*Chakraborty, Moore & Parrillo-Chapman, 2022*).

An enhanced YOLOv5-based technique was used for automated fabric defect detection. A student-teacher architecture was employed to address the limited availability of defected images (*Jin & Niu, 2021*). A deep network, referred to as the teacher network, was specifically designed to predict fabric defects. After refinement, a smaller student network was deployed to perform the same task in real time with minimal performance degradation. Additionally, multitask network learning was introduced to simultaneously detect both pervasive and explicit faults. Principal component analysis and central restrictions were applied to improve detection performance. Evaluations were conducted on the publicly accessible Tianchi AI and TILDA datasets. The results demonstrated that the proposed technique outperformed other approaches and exhibited exceptional fault recognition capabilities in textile images.

Fabric fault recognition is a crucial quality control aspect in the textile industry. A study (*Kumari, Bandara & Dissanayake, 2021*) in 2021 proposed a computer vision-based system utilizing a Sylvester Matrix-Based Resemblance Technique for automated fault detection. The process involved six stages: resolution adjustment, image enhancement using histogram specification and mean median based image cropped histogram equalization, image cataloging using placement and hysteresis procedures, image deletion, edge recognition, and defect recognition using Sylvester matrix ranking (*Kumari, Bandara & Dissanayake, 2021*). The research findings demonstrate the effectiveness and high accuracy of the proposed technique at high computational speeds.

Traditional physical inspection of apparel faces limitations in terms of effectiveness and precision, which can be addressed through image processing and other computer-assisted methods (*Thakur et al., 2023*). Common faults in garment production include sewing faults, seaming faults, positioning faults, pinhole retread, laundry flaws, *etc.* (*Saha et al., 2021*). Manual fault identification is labor-intensive, has low accuracy, and is inefficient. The average accuracy of physical fault recognition is around 60%, and even expert assessors can only detect approximately 70% of faults (*Bangare et al., 2017*). The deep learning concepts have been applied to other domains as well, (*Yu et al., 2024*) have proposed a regression algorithm to detect and identify the defects in printed circuit boards. *López de la Rosa et al. (2023)* used Squeezed Net CNN to detect defects in semiconductor wafers. The study proposed a double staged graphical transformation-based data augmentation strategy. Industrial, manufacturing products defect detection has been remained a hot area. A study has used two stage neural networks for detection and classification of manufacturing products defects (*Avola et al., 2022*). Multistage architectures for detection and classification have more effectiveness but struggles efficiency (*Wang, Fan-Jiang & Lee, 2022*) also have used multi-stage CNN for defect detection and classification for tinny components. CNN and these other techniques are supervised ways of defect detection in deep learning domain. Semi-supervised (*Manivannan, 2023*) and unsupervised (*Zhao et al., 2023*) techniques have also been proposed for defect detection in various industrial problems.

The deep learning practices not only help in detecting the defects from fabric but also have deployed for detection of anomalies in automated fabric manufacturing systems in real time. *Talu, Hanbay & Hatami Varjovi (2022)* proposed CNN based real-time defect detection system for fabric being developed on LOOM. Two distant types of defects namely Warp-Defect and horizontal deformation detect was considered in this study. The research has deployed two different datasets, former containing the real fabric image while later are developed using DPC algorithm. The aim of these datasets to train the CNN for fabric defect detection and classification system. The study has used Fourier analysis to find defect part in fabric video streaming, used effectively for special fabric textures like denim jeans *etc*. Concept of negative mining has been used to improve the dataset for training of. The results of CNN were compared with Fourier analysis shows that CNN model have relatively higher detection and classification accuracy as compared to the traditional computer vision algorithm. This was also due to the use of negative mining of dataset before training of CNN.

*Li & Zhu (2024)* also has proposed a CNN configuration for fabric defect detection and classification. The study claims that the deep learning models are slower while focus on accuracy. The study proposed a faster YOLOv5. The study proposed three changes in the model, firstly PDConv used for quickly pick out important information from image, secondly the enhanced BiFPN used to combine the details from different images. Lastly, the study proposed a better loss function called IN loss which helps the proposed model find small defect and faster learning. The model was trained on a dataset with five common fabric defects namely broken-hole, flower-board, pulp-spot, three-threads, and stain. The proposed changes improved accuracy by 3.6%, reaching 87.9%. It was also tested on another defect dataset (NEU), where it performed better than most existing methods. Finally, they ran the model on a small device (Jetson TX2), and it worked in real time at 31 frames per second, making it suitable for use in factories.

*Liu et al. (2022)* also iterated fabric defects are harder to detect and classify manually with high accuracy due to complex patterns and large variation of defects. Even the computer-based algorithms and models are either slower or less effective. The study proposed improved version of YOLOv4, a very popular detection model already trained with a large amount of data, to detect and localized the fabric defects. The MaxPooling part of YOLOv4 has been replaced with SoftPooling to improve accuracy of the model. The change made the system more effective to understand the underlying features of pertinent part of the image and increase accuracy. The proposed study also has changed the structure of the model to it can accurately process the image information. Adaptive contrast histogram equalization has been used for detection to make image quality better and help YOLOv4 to identify the defects accurately. The experiments shows that these two changes in the structure of YOLOv4 has improve the accuracy of the model up to 6% with more efficiency then earlier.

The You Only Live Once (YOLO) family have effectively been used for defect detection and classification of fabric defects and anomalies. *Guo et al. (2023)* have improved the YOLOv5 real time defect detection and classification in fabric. The new method named AC-YOLOv5 was proposed, which is designed to meet the high demands of fabric defect

detection system. A model containing filters of different sizes are deployed to view the image in multiple ways, helping the model detect defects of different sizes without lowering image quality. Next, another module for convolutional channel attention was added. This helps the model focus on the most important features in the image, making it better at finding defects and more resistant to distractions. The dataset used for experimentation was collected from a circular knitting machine from a production unit. The study has deployed many tests and identify that the improvement of YOLOv5 with additional modules has increased the accuracy of YOLOv5 up to 99.1%, making the YOLOv5 very effective for real-world deployment in fabric manufacturing units.

Another sequel of YOLO family has been used for detection of fabric defects. *Nasim et al. (2024)* have used improve YOLOv8 for detection of fabric defects in real-time production. The study trained a smart model using real factory images from Chenab Textiles. It used a fast and light model called YOLOv8. The results of the customized YOLOv8 were compared with other models like YOLOv5 and MobileNetV2. YOLOv8 gave the best results, with high accuracy and good speed, correctly detecting 7 types of defects listed as de-coloration, stain, grey stitch, baekra, contamination, cut and selvet. *Jin & Niu (2021)* have also improved YOLOv5 for fabric defect detection and classification. Since there were not so many fabric defect images, the study has used a smart trick with two models: a big, strong model teaches a smaller one. The smaller model can then work fast and still give good results. The proposed updated YOLOv5 were tested on two well-known fabric image datasets (Tianchi AI and TILDA). The results showed that this new system works better than many others and can find defects in fabric pictures very well.

*Jia et al. (2022)* have proposed a method to improve the Faster region based convolutional neural network (RCNN) and transfer learning for effective detection fabric defects. In the first step, the study trained a model on ImageNet dataset, later the Improved Faster RCNN was trained on fabric image. The actual part of the model was then replaced with the ResNet50 to avoid mistakes while detecting the region of interest in defect areas. The system has deployed a regional proposal network with a feature pyramid to find possible defects with more accuracy. At the end of the updated Faster RCNN, the SoftMax classifier was deployed. The experiments show that this improved system works better than many other current methods. It is more accurate and learns faster, making it useful for future fabric defect detection.

Another study has also deployed CNN to defect the fabric defects and reduce the false-negative detections (*Almeida, Moutinho & Matos-Carvalho, 2021*). The study claimed that checking fabric for defects is very important for fabric companies. Normally, humans find only 60–75% of defects. To fix this, the study suggests using an automatic system to help find defects quickly. The system proposed a customized CNN which was trained using over 50 types of defects to make it work well on many kinds of problems. The study also highlighted another issue in automated defect detection and classification that sometimes, the system misses defects (called false negatives), which is worse than finding defects that don't exist (false positives). The study proposed two different approaches for false negative reduction; the former is called classification threshold reduction and later was named as rejection region effectively used to reduce false negatives. The experiment

shows that, in automatic mode, the system correctly detects defects about 75% of the time. But if a human helps by checking the results, the accuracy can go up to 95%.

*Carrilho et al. (2024)* has analyzed different automated fabric defect detection methods, focusing on the change from traditional image processing algorithms to advanced machine learning and deep learning techniques. Manual fabric checkups are still extensively used but suffer from low accuracy, higher costing of labor, and inconsistency. The study analyzed multiple traditional approaches including statistical, spectral, model-based, and structural methods, highlighting their pros and cons, especially in detecting fine or rare defects. With the development of deep learning, methods like CNNs and GANs have shown higher accuracy in both detection and classification tasks. The study categorizes detection algorithms into single-stage and two-stage object detectors, explaining their trade-offs in terms of speed and accuracy. The study has reviewed the widely used datasets, noting that many lack diversity or standardization, which hampers reproducibility and fair performance comparison. They emphasize the importance of dataset quality and availability, advocating for broader adoption of newer datasets like ZJU-Leaper. A critical challenge highlighted is the computational demand of deep learning models, making deployment in factory environments difficult without edge-based solutions.

## METHODOLOGY

In this research, we propose a DCGAN model to generate synthetic images of four common defects found in stitched denim jeans: broken loops, broken stitches, twisted leg and skipped stitches. The proposed DCGAN can effectively produce a large quantity of defective images, even with a relatively small preliminary dataset.

### Data acquisition

The original dataset comprised approximately 3,930 images of various denim jeans defects, including 1,300 images of broken loops, 1,100 images of broken stitches, 1,100 images of skipped stitches and 430 images of twisted leg. The images captured denim jeans of different colors and styles, emphasizing dataset diversity. Anonymous industries were visited to collect these images. Many of the defective denim jeans were found in the leftover stock of denim jeans export industries, highlighting the importance of quality assurance in the export industry. The images were captured in red green blue (RGB) format with dimensions of 256 × 256 pixels. Extraneous elements like noise and unnecessary features were removed during preprocessing. Denim jeans are typically assembled from various parts, so non-essential portions of the images were manually removed to obtain a refined dataset.

### Data augmentation

Insufficient datasets can lead to overfitting during the training of deep neural networks, hindering their convergence and effectiveness in generative, detection, and classification models. Given the number of sample images mentioned earlier, it was not suitable for training a DCGAN to achieve optimal results. Data augmentation techniques were employed to expand the dataset.

### Flipping

Flipping was applied both vertically and horizontally. The Python Imaging Library (PIL) was used to invert the pixel values in the images row-wise and column-wise. Both the original and transformed images were saved in a specified output directory.

### Random zooming

Random zooming was applied to generate images with varying zoom levels. The Keras ImageDataGenerator class was used for this purpose. Zooming was performed within a range of 20% to 40% of the original image to ensure that features were not lost. The generated images with different zoom levels were saved in a specified output directory.

### Color space augmentation

To enhance the robustness and diversity of the image dataset, we applied color space augmentation. Using the OpenCV library, we altered the color characteristics of the images by converting them to different color spaces (RGB and HSV) and adjusting brightness, hue, and saturation values within a range of −20% to +20%.

A key observation during the experiment was that rotation, a geometric transformation, can negatively impact the generator model's learning. The generator may prioritize learning the rotation pattern rather than focusing on the underlying probability distribution within the dataset.

### Proposed DCGAN

The architecture of generator and discriminator networks are explained below. Each component in this DCGAN configuration is designed to perform a specific role within the adversarial training framework.

### Generator Network Architecture

The generator model architecture is designed to generate images from simple noise, following a typical DCGAN framework as depicted in Fig. 6. This specialized architecture employs a series of layers that progressively transform random noise into organized, high-dimensional images. Here's a breakdown of the architecture and its significance.

### Input layer

The proposed model begins with a one-dimensional random noise vector of length 100 as input to a dense layer. The generator uses this noise as a latent space to learn how to generate realistic images. The dense layer expands the provided vector to a much larger dimensional space of $8 \times 8 \times 512$. This expansion is crucial for generating high-dimensional images. After the dense layer, batch normalization and rectified linear unit (ReLU) activation are applied. Batch normalization stabilizes the learning process by normalizing each mini-batch input, while ReLU activation introduces non-linearity, enabling the model to learn complex representations.

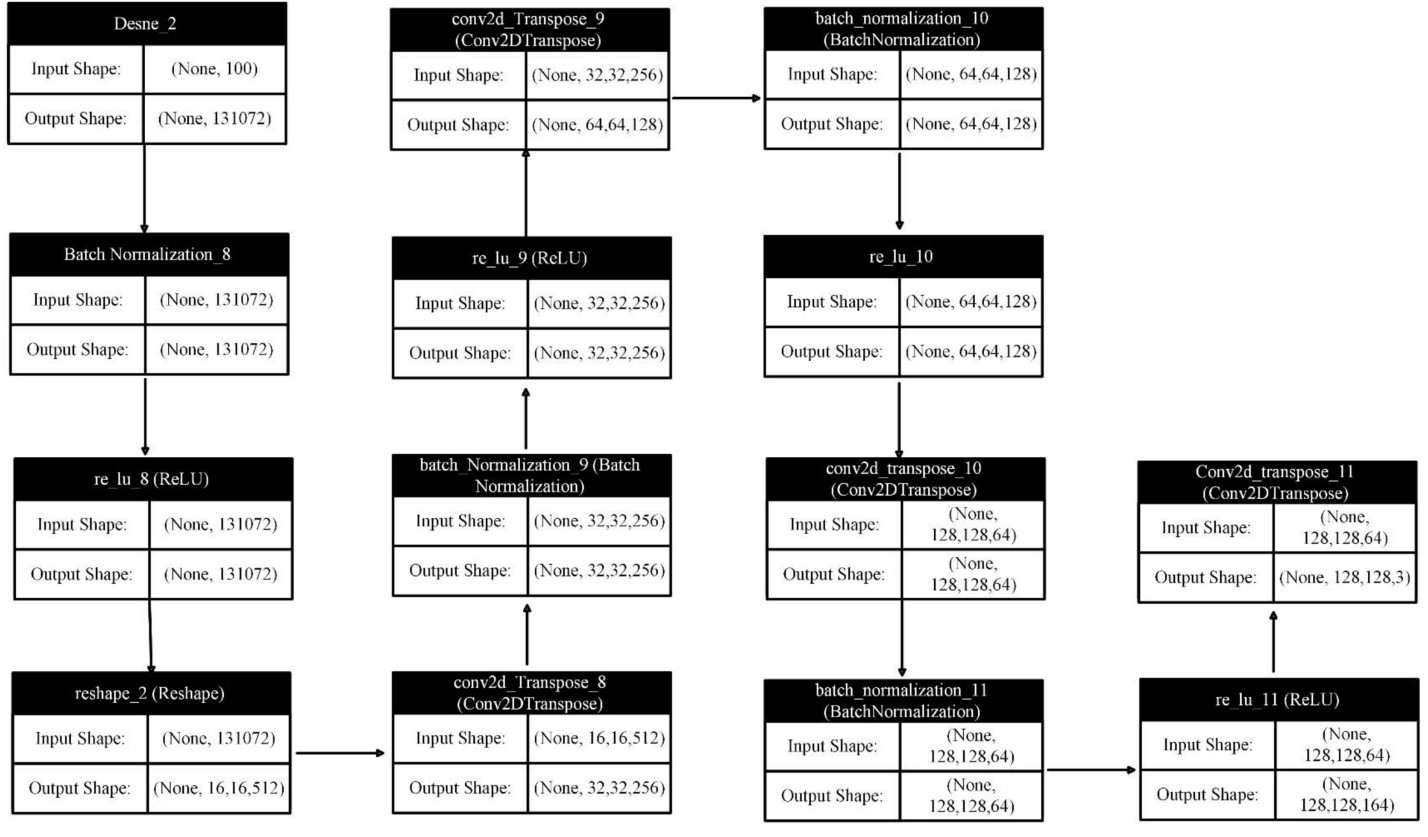

**Figure 6** Generator network layers input and output.

### Reshaping

The output of the dense layer is reshaped into a three-dimensional tensor of size $8 \times 8 \times 512$. This conversion is essential as it transforms the one-dimensional output into a format suitable for the subsequent convolutional layers, which operate in multiple dimensions.

### Upsampling layers

A significant component of the generator network is a series of Conv2DTranspose layers, also known as deconvolutional or upsampling layers. These layers gradually increase the spatial dimensions of the feature maps from $8 \times 8$ to $64 \times 64$, effectively constructing the image layer by layer. To maintain non-linearity and stability, each of the three Conv2DTranspose layers is followed by a batch normalization layer and ReLU activation function. The upsampling process progressively refines the random noise into a structured representation containing the desired image features, adding more detail and complexity at each stage.

### Final layer

The final deconvolutional Conv2DTranspose layer produces an output with the desired dimensions of $64 \times 64 \times 3$. This layer uses a Tanh activation function to generate output

**Table 1  Generator network layers parameters.**

| Layer | Parameters |
|---|---|
| dense_2 (Dense) | 13,238,272 |
| batch_normalization_8 (BatchNormalization) | 524,288 |
| re_lu_8 (ReLU) | 0 |
| reshape_2 (Reshape) | 0 |
| conv2d_transpose_8 (Conv2DTranspose) | 2,097,408 |
| batch_normalization_9 (BatchNormalization) | 1,024 |
| re_lu_9 (ReLU) | 0 |
| conv2d_transpose_9 (Conv2DTranspose) | 524,416 |
| batch_normalization_10 (BatchNormalization) | 512 |
| re_lu_10 (ReLU) | 0 |
| conv2d_transpose_10 (Conv2DTranspose) | 13,136 |
| batch_normalization_11 (BatchNormalization) | 256 |
| re_lu_11 (ReLU) | 0 |
| conv2d_transpose_11 (Conv2DTranspose) | 3,075 |

**Note:**
Total params: 16,520,387 (63.02 MB), Trainable params: 16,257,347 (62.02 MB), Non-trainable params: 263,040 (1.00 MB).

values within the range of −1 to 1, a common practice in GANs to ensure that the pixel values of the generated images fall within the same range as those of the real images used during training.

This generator design is a structured pipeline for generating images from a simple noise vector. The initial dense layers start the training process, followed by a reshaping layer and upsampling using a series of Conv2DTranspose layers. This architecture enables the model to begin with random noise and progressively refine it into a higher-dimensional image. To stabilize training, each Conv2DTranspose layer is supported by batch normalization and ReLU activation functions. This combination empowers the model to learn complex transformations essential for producing realistic images. The final layer uses a Tanh activation function to ensure the synthetic images contain pixel values suitable for subsequent processing. This architecture's ability to learn complex distributions and generate highly detailed images from noise is attributed to this combination of components.

The number of trainable parameters in each layer of the generator network has been shown in Table 1. Moreover, the layer architecture of generator network is depicted in Fig. 7.

The input to the generator be a latent vector $z \in R^{(LATENT_{DIM})}$. The generator $G(z)$ in is a function mapping the latent space $z$ to an image in pixel space $G : R^{(LATENT_{DIM})} \rightarrow R^{(H \times W \times CHANNELS)}$. The generator consists of several layers, each of which applies a transformation. In Eq. (4) We define the output $G(z)$ as a composition of several functions $T_i$ that represent transformations through the network's layers:

$$G(z) = T_n \circ T_{(n-1)} \circ \cdots \circ T_1(z), \tag{4}$$

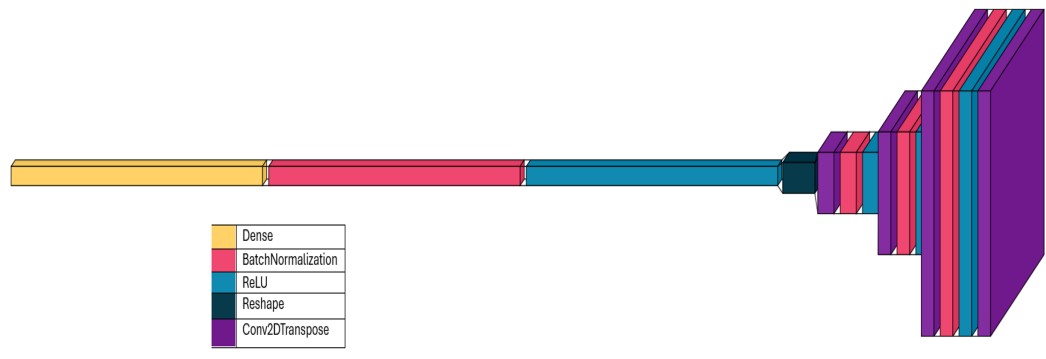

**Figure 7 Layer architecture in generator network.**

where, ∘ denotes the composition of functions, and each $T_i$ represents a specific layer transformation. At each layer, the transformation can be viewed as a differential operator applied to the input tensor (feature map) from the previous layer. Transformation at layer $i$ can be seen in Eq. 5.

$$T_i(h_{(i-1)}) = \sigma(\partial/(\partial h_{(i-1)}) \int_{(V_i)} \text{▓} W_i \cdot h_{(i-1)} dV + b_i), \tag{5}$$

where $h_{(i-1)}$ is the input feature map to the layer. $W_i$ and $b_i$ denotes the weights and biases at layer $i$. The activation, ReLU/Tanh function is expressed as $\sigma$. The term $\partial/(\partial h_{(i-1)})$ denotes the gradient of the layer output with respect to theinput, and similarly, $\int_{(V_i)} \text{▓} W_i \cdot h_{(i-1)}$ represents the convolutional or fully connected operation in terms of a continuous volume integral over $V_i$, the receptive field of the layer.

Suppose, for the first dense (fully connected) layer, the output is given in Eq. 6:

$$h_1 = ''ReLU''(\partial/\partial z \int_{(V_1)} \text{▓} W_1 \cdot z dV + b_1), \tag{6}$$

where $W_1$ represent the weight matrix for the dense layer and $b_1$ denotes the bias term. Each transposed convolution layer performs an upsampling operation. Equation (7) shows the upsampling from $16 \times 16$ to $32 \times 32$. Where $V_i$ represents the volume of the convolution kernel, and the operation is applied elementwise across the feature map.

$$h_3 = ''ReLU''(\partial/(\partial h_2) \int_{(V_3)} \text{▓} W_3 \cdot h_2 dV + b_3) \tag{7}$$

The final output layer applies a tanh activation to map the values to the range $[-1, 1]$. The fake sample is mathematically expressed in Eq. (8).

$$x_{''fake''} = \tanh(\partial/(\partial h_{(n-1)}) \int_{(V_n)} \text{▓} W_n \cdot h_{(n-1)} dV + b_n), \tag{8}$$

where $x_{''fake''} \in R^{(64 * 64 * CHANNELS)}$ represents the generated image.

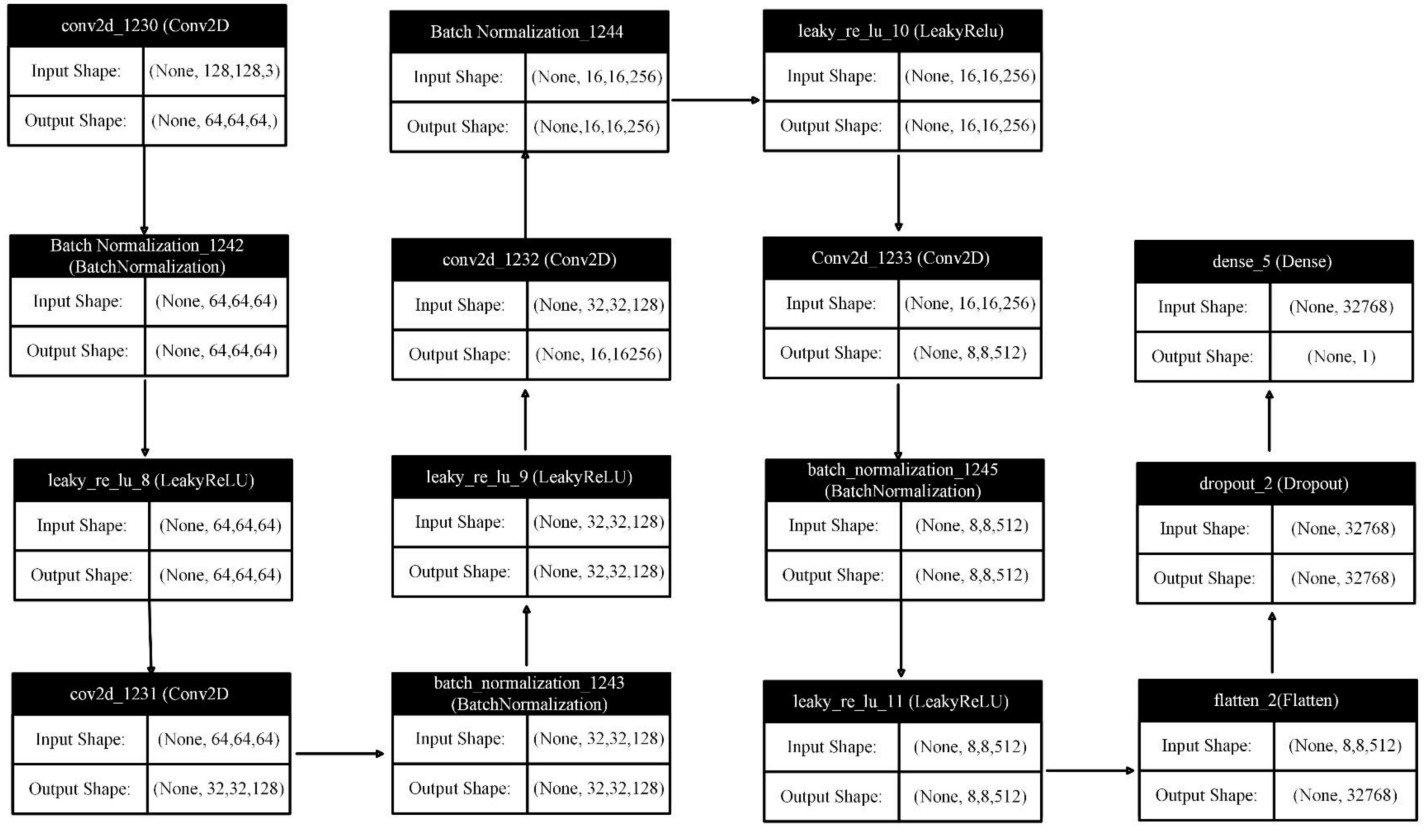

**Figure 8 Discriminator network layers input & output.**     

The overall generator function can be seen in Eq. (9) expressed as a nested composition of transformations:

$$G(z) = \tanh [\![(\partial/(\partial h_{(n-1)}) \int_{(V_n)} \vdots W_n \cdot (\sigma(\partial/\partial z \int_{(V_1)} \vdots W_1 \cdot z dV + b_1))dV + b_n)]\!]. \qquad (9)$$

## Discriminator network architecture

The discriminator network is designed with an aim to distinguish between real and synthesized image instances. The discriminator model proposed in the research was created using the Sequential API from Keras, allowing layers to be stacked sequentially. An image of size 128 × 128 with 3 RGB color channels was provided as input to this model. The discriminator input and output network layer details are depicted in Fig. 8.

## Convolutional layers

The discriminator model consists of four Conv2D convolutional layers, each combined with a batch normalization layer and Leaky ReLU activation layer. Each convolution has a kernel size of (4, 4) and a stride of (2, 2), effectively reducing the image's spatial dimensions by half in each layer. The filter size increases in successive convolutional layers, starting

from 64 and progressing to 128, 256, and finally 512. This allows the model to learn increasingly complex features at various levels of abstraction.

## Batch normalization

To stabilize and accelerate the training process, each convolutional layer was followed by a batch normalization layer. These layers regulate the output of the preceding activation layer, helping to prevent internal covariate shift issues.

## Leaky ReLU activation

The LeakyReLU activation function was used with an alpha value of 0.2. Unlike the standard ReLU, which sets all negative inputs to zero, Leaky ReLU allows a small gradient for negative inputs. This helps to prevent neurons from becoming inactive during training.

## Flattening and dropout

A Flatten layer follows the convolutional layers, flattening the output into a single vector. This conversion is necessary for the subsequent dense layer. After flattening, a dropout layer with a 0.3 dropout rate is applied to prevent overfitting by randomly setting a portion of input units to zero during training.

## Output layer

The final layer is a dense layer with a single neuron and a sigmoid activation function. This dense layer produces a probability score between 0 and 1, representing the likelihood that the provided image is real. A score closer to 1 indicates a real image, while a score closer to zero suggests a synthetically generated image.

The input to the discriminator be an image $x \in R^{(64 * 64 * 3)}$. The discriminator $D(x)$ as expressed in Eq. (10) is a function mapping the image space to a probability value indicating whether the input is real or fake. The discriminator consists of several layers, each of which applies a transformation. We define the output $D(x)$ as a composition of several functions $T_i$ that represent transformations through the network's layers.

$$D(x) = T_5 \circ T_4 \circ T_3 \circ T_2 \circ T_1(x), \tag{10}$$

where "$\circ$" denotes the composition of functions, and each $T_i$ represents a specific layer transformation. At each layer, the transformation can be viewed as a differential operator applied to the input tensor from the previous layer. The transformation at layer $i$ is expressed in Eq. (11).

$$\mathrm{Ti}(hi-1) = \sigma(\partial/(hi - 1\partial) \int \llbracket V_i W_i \rrbracket \cdot h_i - 1 dV + b_i), \tag{11}$$

where $h_{(i-1)}$ is the input feature map to the layer. $W_i$ and $b_i$ denotes the weights and biases at layer $i$. LeakyReLU activation function is represented as $\sigma$. The Term $\partial/(\partial h_{(i-1)})$ denotes the gradient of the layer output with respect to the input and $\int \llbracket V_i W_i \rrbracket \cdot h_i - 1 dV$ represents the convolutional operation over the receptive field $V_i$.

Suppose for the first convolutional layer, the output is expressed in Eq. (12).

**Table 2 Shapes and parameters detail in discriminator network.**

| Layer | Parameters |
|---|---|
| conv2d_1230 (Conv2D) | 3,136 |
| batch_normalization_1242 (BatchNormalization) | 256 |
| leaky_re_lu_8 (LeakyReLU) | 0 |
| conv2d_1231 (Conv2D) | 131,200 |
| batch_normalization_1243 (BatchNormalization) | 512 |
| leaky_re_lu_9 (LeakyReLU) | 0 |
| conv2d_1232 (Conv2D) | 524,544 |
| batch_normalization_1244 (BatchNormalization) | 1024 |
| leaky_re_lu_10 (LeakyReLU) | 0 |
| conv2d_1233 (Conv2D) | 2,097,664 |
| batch_normalization_1245 (BatchNormalization) | 2,048 |
| leaky_re_lu_11 (LeakyReLU) | 0 |
| flatten_2 (Flatten) | 0 |
| dropout_2 (Dropout) | 0 |
| dense_5 (Dense) | 32,769 |

**Note:**
Total params: 2,793,153 (10.66 MB), Trainable params: 2,791,233 (10.65 MB), Non-trainable params: 1,920 (7.50 KB).

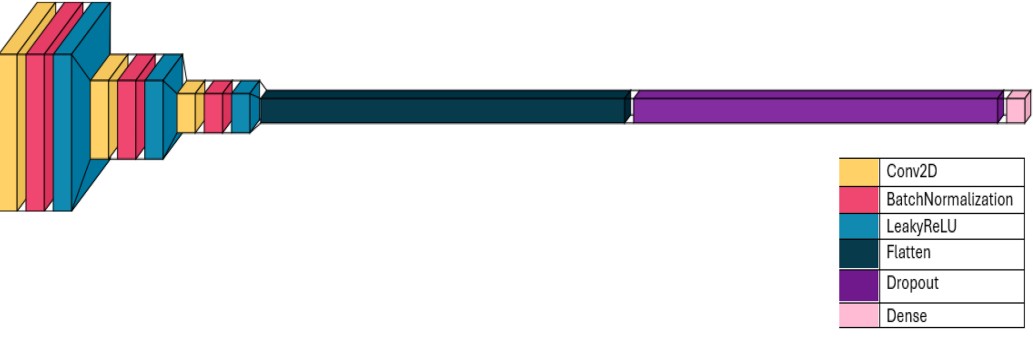

**Figure 9 Layer architecture in discriminator network.**

$$h_1 = ''LeakyReLU''(\partial/\partial x \int_{(V_1)} W_1 \cdot x dV + b_1) \tag{12}$$

Thus, the overall discriminator function can be seen in the Eq. (13) compactly expressed as a nested composition of transformations:

$$D(x) = \sigma(\partial/(\partial h_4) \int_{(V_4)} W_4 \cdot (''LeakyReLU''(\partial/\partial x \int_{(V_1)} W_1 \cdot x dV + b_1))V + b_n). \tag{13}$$

The discriminator model is designed as a binary classifier capable of distinguishing between real and generated images. The use of convolutional layers allows it to learn spatial characteristics within images, while batch normalization and dropout layers contribute to a robust and stable training process. The Leaky ReLU activation function helps to mitigate

**Table 3  Parameters of training.**

| Parameters | Value |
| --- | --- |
| Image scale | [−1, 1] |
| Optimizer adam | Adam |
| Learning rate generator | 0.0001 |
| Learning rate discriminator | 0.0003 |
| Momentum rate | 0.5 |
| Training iterations | 2,000 epochs |
| Dropout | 0.3 |
| Latent space dimension (z) | 100 |
| Normalization | Batch normalization |
| Loss function | BinaryCrossEntropy |

the dying ReLU problem, ensuring that the model learns effectively. The number of trainable parameters in each layer of the network has been shown in Table 2. Moreover, the layer architecture of discriminator network is depicted in Fig. 9.

## Training

During the training, the generator network is evaluated using a loss function $V(G)$, defined in Eq. (1). It takes random noise z (a 100-dimensional vector) as input and generates data $G(z)$, which is then assessed by the discriminator. The discriminator tried to distinguish between real data x and generated data $G(z)$, using its own loss function $V(D)$, defined in Eq. (2). During training, $G$ tried to minimize $V(G)$, effectively aiming to maximize $D$'s error, while $D$ minimizes $V(D)$ to improve its classification accuracy. This adversarial process helped both networks improve iteratively, with $G$ learning to generate increasingly realistic data and $D$ enhancing its ability to distinguish real from fake. Table 3 provides a list of parameters set for training the generator and discriminator networks.

The code was developed in a Kaggle notebook and executed on an Nvidia Tesla P100 GPU with 6.0 Compute Capability. Kaggle notebooks provide a powerful environment for deep learning projects, featuring dual Intel Xeon processors and 13 GB of RAM. Kaggle notebooks support a broad range of CUDA operations, significantly accelerating the training process for models like DCGAN. The Adam optimizer was employed to train both the generator and discriminator networks. Different learning rates were carefully selected for the generator and discriminator.

Various technical aspects that make the model different from the standard DCGAN architecture as proposed by *Radford, Metz & Chintala (2016)*. The standard GAN typically outputs 64 × 64 images from a lower-resolution images, while the proposed generator begins with the dense extension to 8 × 8 × 512 tensor. This modification empowers the model with a deeper and more elaborative latent representation early in the training. A broad up sampling approach using many Conv2DTranspose layers progressively raise the image dimension to 64 × 64 having ReLU activation functions in each layer and batch normalization after each layer. This strategy ensures stable training and better convergence. The proposed architecture used Tanh activation function in the output layer,

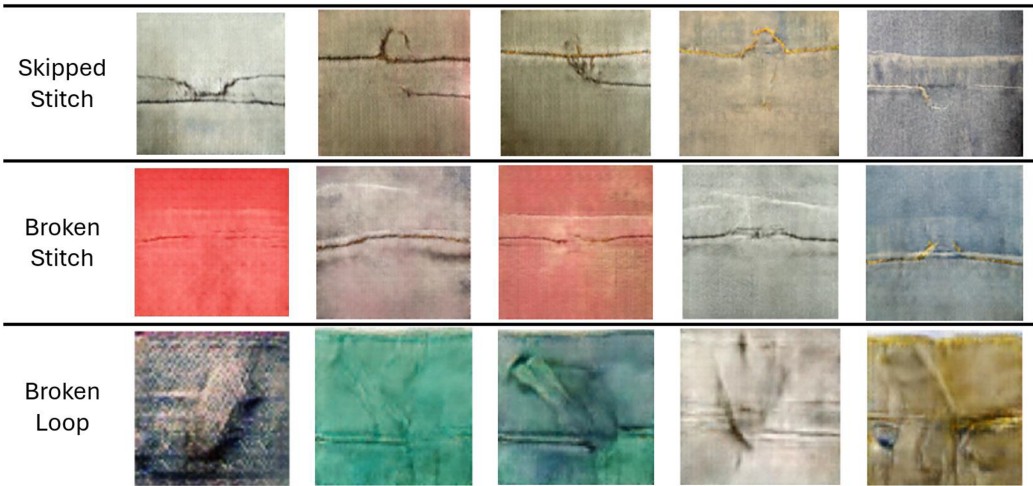

**Figure 10 Broken loop defect images generated by DCGAN for 128 × 128 pixel window size.**

this change enables the creation of image with standard pixel intensities. The discriminator network was also improved to extract gradually intricate feature representations through four convolutional layers. Each convolutional layer in discriminator network expands the feature map from 64 to 512 filters. The gradient vanishing problem has been mitigated using the LeakyReLU activation function throughout the architecture. Moreover, the proposed DCGAN was improved specifically for the area of denim jeans defect synthesis by training on real-world denim defect patterns.

The proposed DCGAN has been compared to another similar model (*ul-Huda et al., 2024*) developed for the synthesizing the fabric defect images. The proposed model achieves superior performance as discussed in the later section as compared to reference model with a notably simplified network having a smaller number of layers in both generator and discriminator. The reference model has employed multiple convolutional and up sampling blocks, our proposed generator network used a fewer conv2Dtranspose layers and starts with a more expressive dense mapping (8 × 8 × 512), tailed by an efficient upsampling training with ReLU activations. Similarly, the discriminator network in the proposed model is also using less convolutional layer than the referenced model but retains the essential components like activation functions, normalization, and dropout stability. This thin arrangement decreases parameter count and reduces the computational cost though synthesizing the better-quality results, as demonstrated by our better performance scores. The proposed model is a more efficient adversarial training cycle with faster convergence and better performance.

## Experimental results and discussion

The DCGAN model was trained on a dataset comprising approximately 3,930 seed and similar number of augmented images, representing four distinct defect types: broken loops, broken stitches, skipped stitches, and twisted leg. These defects are commonly

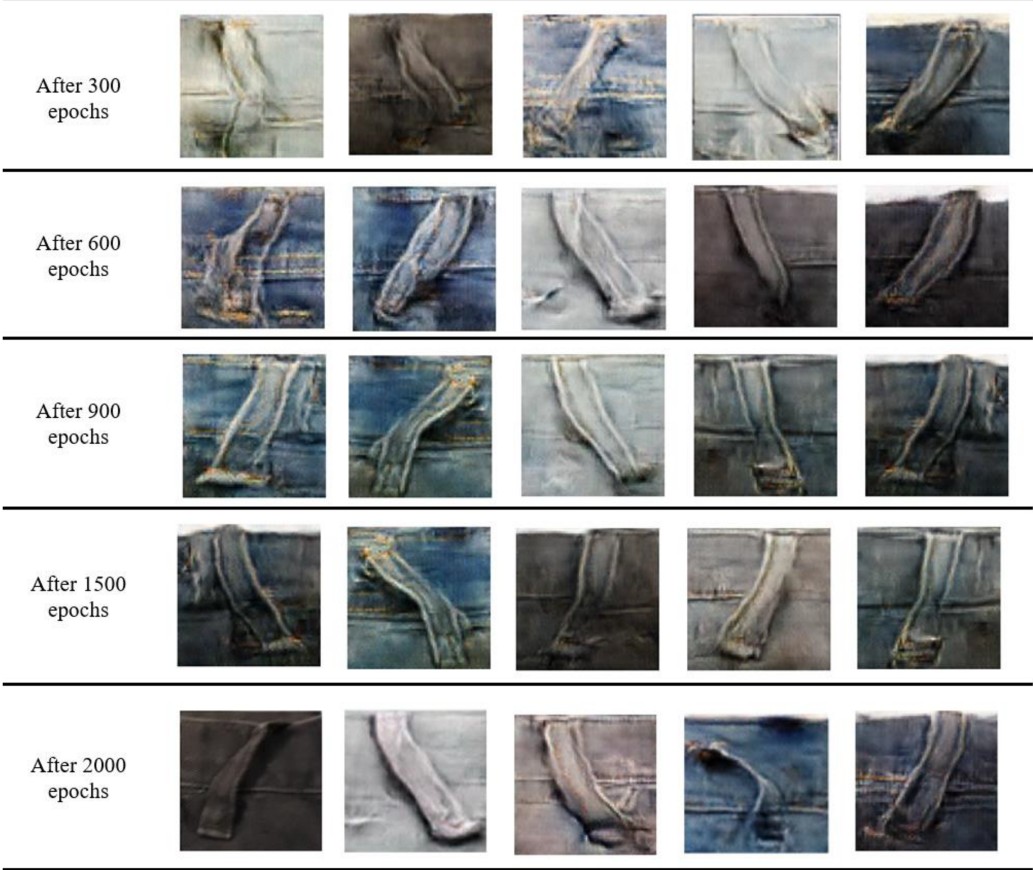

**Figure 11 Broken loop defect images generated by DCGAN for 256 × 256 pixel window size.**

observed in stitched denim jeans. The dataset also included augmented images, as described earlier. To enhance training efficiency, each model was trained separately for each defect type.

Initially, the model was deployed using a 128 × 128 pixel window size. However, even after 2,000 training epochs, the results were unsatisfactory, as shown in Fig. 10. In the second iteration, all images were resized to 256 × 256 pixels. The architecture of the network has already been explained in the 'Results' section. Training again lasted for 2,000 epochs, with the model saving output after each epoch. A selection of defect samples generated during different epochs is presented in this study for reference.

### Broken loop defect

Figure 11 illustrates the images of broken loop defects generated by the DCGAN across various epochs. In the initial training stages, the model starts learning the underlying distribution of the provided dataset. After 300 epochs, it begins to produce denim jeans loop-like shapes, although the images remain blurry and noisy. By 600 epochs, the generator has improved its feature learning, starting to generate recognizable

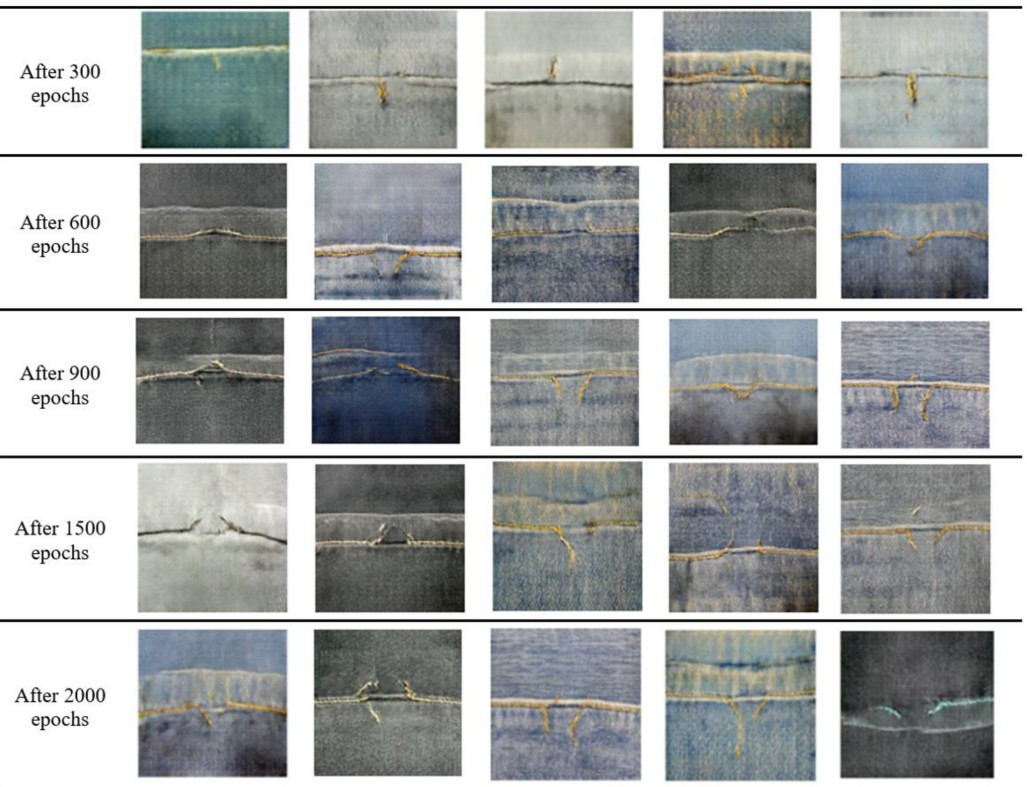

**Figure 12 Broken stitch defect images generated by the DCGAN for 256 × 256 pixel window size.**

broken loops of denim jeans, albeit still fragmented. As training progresses to 900 epochs, the generated images become clearer and more closely resemble the real dataset. The training process continued to produce increasingly realistic broken loop images, reaching near-perfection at around 1,500 epochs. Ultimately, by deceiving the discriminator, the generator successfully generated images similar to those provided for training.

## Broken stitched defect

Figure 12 presents the images of broken stitch defects generated by the proposed model at various training stages. Initially, the model begins to grasp the underlying distribution of broken stitch defects, as evident at 300 epochs. While broken stitch defect images are produced, they initially appear unclear. By 600 epochs, the generator's learning has improved, resulting in visible stitches in the images. Reaching 900 epochs, the model has learned the underlying hierarchies in the training dataset, generating images with significant progress. By minimizing the generator's loss, the model has produced images that closely resemble the original images near 2,000 epochs. The generated images by the proposed DCGAN effectively mimic real defect images. This model can contribute to the creation of a large dataset of defect images.

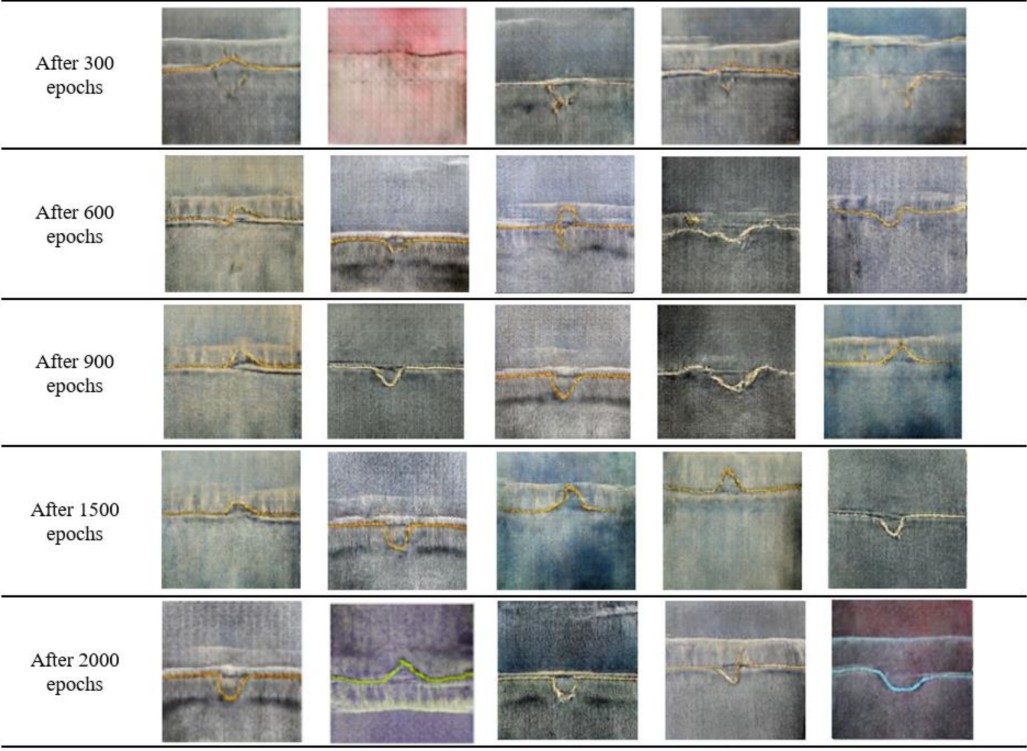

**Figure 13 Skipped stitch defect images generated by DCGAN.**

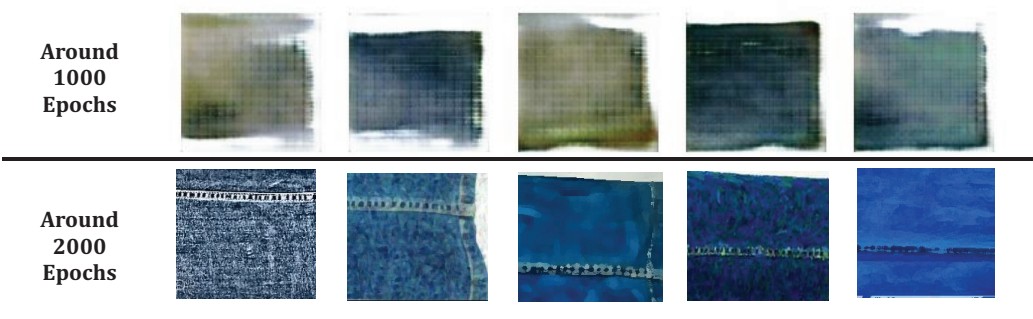

**Figure 14 Twisted leg defect images generated by DCGAN.**

## Skipped stitch defect

Figure 13 shows the images of skipped stitch defects generated by the proposed DCGAN during the training process. Skipped stitches and broken stitches are similar defects, which explains why the generator initially focused on generating stitches on denim jeans. After 600 epochs, the generator began producing recognizable skipped stitch images. Figure 13 illustrates the results of the training process at different stages. Ultimately, the generator

**Table 4 Defect-wise evaluation by industrial expert (out of 100).**

| Industrial expert | Score on defect category in percentage | | | |
|---|---|---|---|---|
| | Broken loop | Broken stitches | Skipped stitches | Twisted leg |
| 1 | 75 | 85 | 85 | 85 |
| 2 | 85 | 90 | 95 | 80 |
| 3 | 85 | 95 | 90 | 75 |
| 4 | 80 | 90 | 90 | 85 |
| 5 | 80 | 85 | 85 | 85 |
| 6 | 50 | 70 | 70 | 85 |
| 7 | 80 | 85 | 95 | 80 |
| 8 | 50 | 70 | 65 | 85 |
| 9 | 75 | 95 | 85 | 75 |
| 10 | 85 | 90 | 90 | 75 |

started generating images that were indistinguishable from real images. The model successfully learned the complex representations contained in the input images, and the generator minimized the training loss, deceiving the discriminator network.

## Twisted leg

The images of twisted leg defect in denim jeans generated by the proposed DCGAN during and after the training are presented in Fig. 14. The seed data was not sufficient to produce the desired visual quality of the defect samples. However, the generator has synthesized the recognizable instances of twisted legs. Figure 14 demonstrates the image samples synthesized at different epochs.

## Evaluation

The performance evaluation of the DCGAN was divided into two phases. The generated images were inspected visually to assess the model's performance. Additionally, the distance between generated and actual images was calculated as another evaluation metric. This section presents the evaluation of the proposed DCGAN using both methods.

## Expert judgement

Expert judgment is a subjective method for performance evaluation and results validation. For evaluation, the generated samples were shared with apparel industry professionals. Eight out of ten professionals agreed that the generated samples, especially broken stitches and skipped stitches, and generally broken loops, appeared authentic. They endorsed these samples for training deep learning models for automatic defect detection and classification. The reviewers also confirmed that the generated images contained the characteristic features of the original defective images. The quantitative scores by different domain and industrial experts have been demonstrated in Table 4. The scores assigned by the evaluators are presented in the graph in Fig. 15. By aggregating these scores, an impressive accuracy of 81.5% was achieved, demonstrating remarkable success. The generated images effectively replicated the features of actual denim jeans defects.

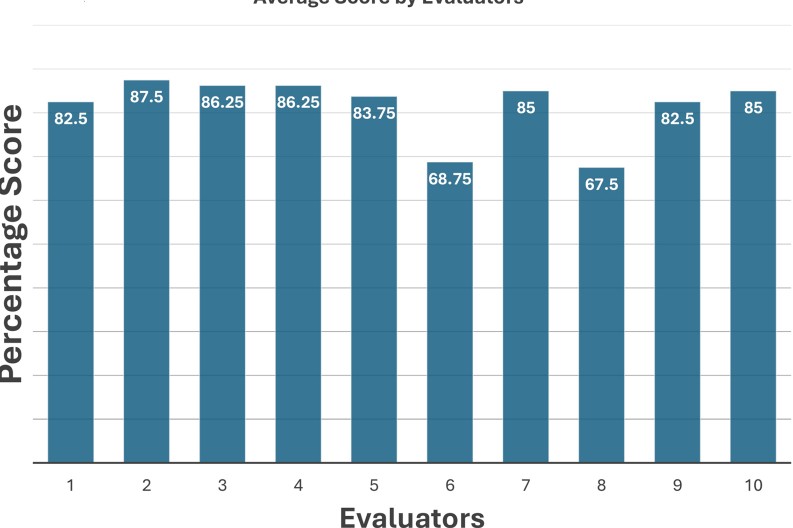

**Figure 15  Average score by ten industrial experts.**

**Table 5  FID score for different defects.**

| Type of defect | FID (Frechet inception distance) |
| --- | --- |
| Broken loop | 12.26 |
| Broken stitch | 6.75 |
| Skipped stitch | 7.68 |
| Twisted leg | 27.59 |

## Objective evaluation

FID is a quantitative metric for calculating the distance between feature vectors of real and synthetically generated images (*Yu, Zhang & Deng, n.d.*). This metric measures the similarity between groups of images in the context of computer vision. A lower FID score indicates greater similarity between the groups of images or similar features. Identical groups would have an FID score of 0.0, while completely different image groups would have an FID score of 100. The FID metric is commonly used to assess the performance of GAN models (*Kumar & Singh, 2024*). It utilizes the InceptionV3 model to extract features and then calculates the score based on the means and covariances of the feature representations.

Table 5 presents the FID scores for all four types of defects, calculated using Keras's applications. inception_v3 library.

The results demonstrate that the DCGAN performed exceptionally well on the broken stitch defect, achieving the lowest FID score of 6.75. This indicates that the DCGAN produced images highly similar to the real dataset. The skipped stitch defect had a slightly higher FID score of 7.68, still demonstrating excellent performance by the proposed DCGAN. The broken loop defect proved to be the most challenging, with more complexity and features in the images, resulting in the highest FID score of 12.26. The twisted leg

defect have highest FID score of 27.59 due to shorter seed size. This suggests slightly less accurate image production for this defect type. The FID score measures the distance between the generated and real image groups. As a significant portion of the images consist of denim jeans, a small area indicates the defect. The pixel-wise calculation of the FID score might not accurately reflect the accuracy of the generated defective portion.

## Parameter analysis for FID score optimization

The impact of various training parameters on the quality of synthetically generated denim defect images using the proposed DCGAN architecture is analysed in this section. FID scores have been used to assess the visual quality of generated images across different parameter configurations. Each parameter was varied individually while keeping the remaining settings consistent with the configuration that yielded the best FID score. The model was trained for up to 2,000 epochs, and FID scores were recorded at regular intervals to monitor convergence behaviour. A consistent improvement in FID scores was observed as the number of training epochs increased, indicating that extended training enables the generator to capture more complex defect patterns. As previously discussed in the 'Introduction', smaller window sizes such as $128 \times 128$ did not produce satisfactory results. Larger window sizes, particularly $256 \times 256$, significantly improved FID scores across all defect types, suggesting that higher-resolution inputs help the model capture finer defect features. The learning rate of both the generator and discriminator networks was also found to have a substantial effect. A lower learning rate for the generator improved image quality, likely because it facilitated more stable and gradual learning, thereby reducing the risk of mode collapse. Similarly, decreasing the discriminator's learning rate led to better FID scores, likely by allowing the generator-discriminator dynamics to remain balanced. In addition, batch size was found to play a critical role in training stability and convergence. Larger batch sizes resulted in lower FID scores, likely due to smoother gradient estimates and improved generalization. Overall, the results presented in Table 6 comprehensively summarize the effects of training epochs, window size, learning rate, and batch size on the FID scores. These findings demonstrate that careful tuning of these hyperparameters is essential for generating diverse and high-quality synthetic defect images. The optimal training parameters have been shown in Table 3. These settings consistently yield the lowest FID scores, indicating high-quality defect generation.

## Impact of synthetic data generation on detection performance

To evaluate the impact of synthetically generated denim defect images produced by the proposed DCGAN architecture, we conducted a comprehensive set of experiments using the YOLOv8 detection model. The goal of the experimentation was to assess whether the augmented dataset with DCGAN generated dataset improve the detection and classification ability of the model. This section presents the dataset configuration, training setup, performance evaluation and visualization of detection outcomes.

The baseline dataset used in this experiment comprised of denim jeans defects captured in realistic denim jeans manufacturing units. Each image in the dataset was annotated with

Table 6 Parametric analysis for FID score optimization.

| Parameter | Values | FID | | | |
|-----------|--------|-----|-----|-----|-----|
| | | Broken loop | Broken stitch | Skipped stitch | Twisted leg |
| Epochs | 300 | 49.57 | 43.58 | 44.48 | 76.58 |
| | 600 | 35.95 | 28.74 | 31.82 | 58.61 |
| | 900 | 24.51 | 18.93 | 21.35 | 45.27 |
| | 1,500 | 18.89 | 10.47 | 12.69 | 36.48 |
| | 2,000 | 12.26 | 6.75 | 7.68 | 27.59 |
| Window size | 128 × 128 | 67.57 | 47.58 | 49.48 | 76.58 |
| | 256 × 256 | 12.26 | 6.75 | 7.68 | 27.59 |
| Learning rate (Generator) | 0.005 | 45.95 | 28.74 | 31.82 | 58.61 |
| | 0.001 | 34.51 | 18.93 | 21.35 | 45.27 |
| | 0.0005 | 18.89 | 10.47 | 12.69 | 36.48 |
| | 0.0001 | 12.26 | 6.75 | 7.68 | 27.59 |
| Learning rate (Discriminator) | 0.005 | 46.83 | 30.24 | 33.12 | 58.61 |
| | 0.001 | 36.39 | 19.87 | 23.45 | 45.27 |
| | 0.0006 | 22.54 | 11.64 | 13.28 | 36.48 |
| | 0.0003 | 12.26 | 6.75 | 7.68 | 27.59 |
| Batch size | 8 | 26.75 | 15.82 | 17.93 | 39.47 |
| | 16 | 18.51 | 9.24 | 11.25 | 31.32 |
| | 32 | 12.26 | 6.75 | 7.68 | 27.59 |

bounding boxes and corresponding defect labels including already stated defect types *i.e.*, broken stitch, skipped stitch, broken loop, and twisted leg.

To address the data scarcity and class imbalance, we have generated additional samples of specified defect regions using the proposed DCGAN for each defect type.

Concisely, we used two different training datasets for comparative evaluation:

Dataset-1: contains only the real annotated images of denim jeans defect specific regions

Dataset-2: contains all images of Dataset-1 alongside the synthetically generated images produced using proposed DCGAN architecture.

Additionally, a separate test set comprising of 200 real unseen complete denim jeans image with no, one or multiple defects in every denim jean image was used for comparative evaluation.

## Experiment

The evaluation was performed using YOLOv8, which was trained from scratch with no preload weights to isolate the impact of data augmentation. Tesla T4 GPU in Kaggle was used for experimentation; the key training parameters were as follows:

Image size: 640 × 640 pixels (input resized)

Batch size: 16

Epochs: 4

Learning rate: 0.001

Optimizer: Stochastic Gradient Descent (SGD)

**Table 7 Performance comparison of YOLOv8 on real and augmented datasets.**

| Defect type | Precision | Recall | F1-score | support | Precision | Recall | F1-score | support |
|---|---|---|---|---|---|---|---|---|
| | Dataset-1 (Real only) | | | | Dataset-2 (Real + Synthetic) | | | |
| BrokenLoop | 0.72 | 0.66 | 0.69 | 220 | 0.79 | 0.75 | 0.77 | 217 |
| BrokenStitch | 0.65 | 0.73 | 0.69 | 128 | 0.81 | 0.84 | 0.82 | 127 |
| SkippedStitch | 0.68 | 0.76 | 0.72 | 130 | 0.78 | 0.81 | 0.79 | 127 |
| TwisedLeg | 0.84 | 0.82 | 0.83 | 85 | 0.81 | 0.87 | 0.84 | 87 |
| Accuracy | 72.32 | | | | 80.28 | | | |
| mAP | 0.72 | | | | 0.8 | | | |

Loss function: Combined objectness, classification, and IoU loss (as per YOLOv8 default)

The model was trained independently on both Dataset-1 and Dataset-2 under the same experimental conditions and evaluated on same standard object detection metrics *i.e.*, precision, recall, F1-score, and mean average precision (mAP) as these metrices provide a comprehensive insight of model's accuracy.

# RESULTS

The comparison between the performance of YOLO on Dataset-1 and Dataset-2 is presented in Table 7. The model trained on the augmented dataset, *i.e.*, Dataset-2, has shown a substantial improvement across all evaluation measures.

A considerable gain in precision, recall and F1-score indicate that the addition of synthetically generated images in a real dataset detect more instances correctly while also avoiding overfitting of the model on real training data. Class wise analysis also shows that the data generated through DCGAN have improved detection as compared to the baseline dataset. Figure 16 shows the samples of defects accurately detected and localized by the YOLO model trained with the augmented dataset.

The detection results confirmed that the synthetically generated images using the proposed DCGAN improve the model's detection accuracy as well as the model's generalization. By synthesizing the additional training instances and intra-class diversity, the Dataset-2 mitigated the possibility of overfitting and increased the robustness of the model.

## Generalization of proposed DCGAN

To assess the generalizability of the model, the proposed DCGAN was trained on a publicly available anime dataset containing 20,589 images (Anime Dataset 2025, https://www.kaggle.com/datasets/monafatima7091/animationdataset). The model was trained for 200 epochs, and the results are presented in Fig. 17. The experiment achieved a FID score of 14.87, indicating high-quality image generation. The following generated images ensuring the robustness of the proposed model.

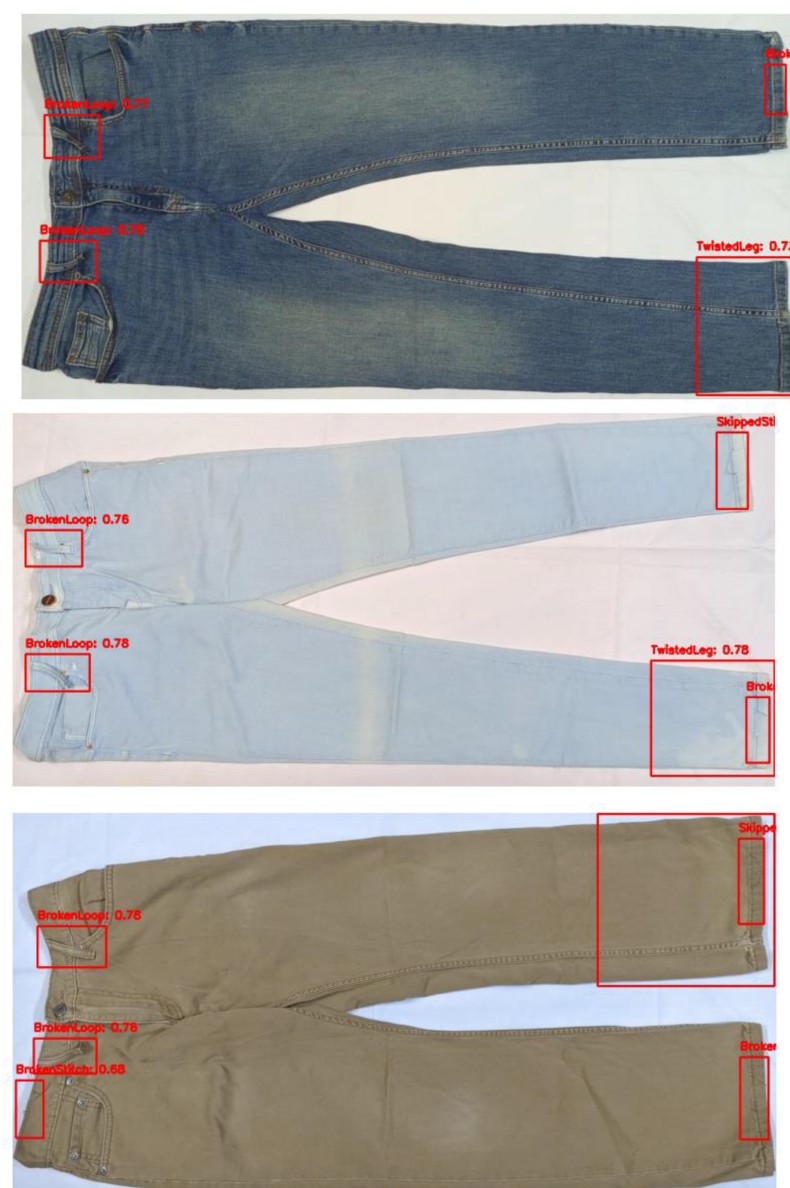

**Figure 16 Skipped stitch defect images generated by DCGAN.**

## DISCUSSION

In the recent surge of industry 4.0, researchers are more focused on providing optimized, intelligent and smart solutions for industrial problems that would increase the efficiency and effectiveness of the industrial processes including production and quality control. In particular, in the textile industry, quality control is an essential part of supply chain. During quality inspection, defects get prime importance as they may lead to order rejection and subsequent financial loss along with the goodwill of the production industry. However, the industry, particularly denim jeans manufacturers, are currently using traditional

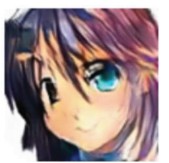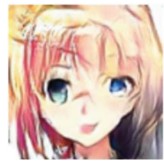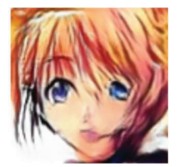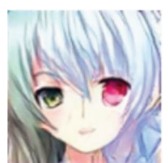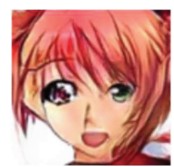
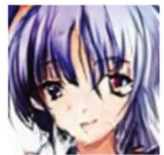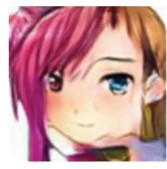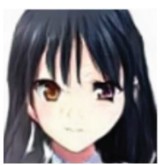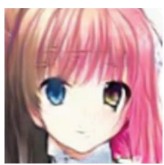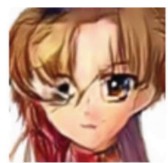

**Figure 17 Image generation for different domain using the proposed DCGAN to validate generalizability.** Images from Anime Dataset 2025 (https://www.kaggle.com/datasets/monafatima7091/animationdataset).

manual inspection mechanism for defect detection that is not only tedious job but may also lead to subsequent issues in quality control. Among several other factors, less trained labor having limited education with a lack of interest and intensive duty hours are some of reasons that may lead to improper quality inspection and may not detect the defects appropriately. Therefore, alternative optimized and smart solutions are required for the underlying task. Artificial intelligence, especially machine and deep learning algorithms, may be trained and implanted in the embedded systems that may detect the defects efficiently and effectively. However, such algorithms are data hungry and need enough data to learn the distribution of defects. Since data for defects are not usually gathered or stored in industrial units. Since standard data is required for the training of algorithms, and only trained professionals may capture such visual data in proper lights and dimensions. Moreover, remote industrial units also become hurdles to gather enough data from the multiple manufacturing unit to provide robust solutions that may be generalized enough to work for all. However, industries do not usually share defects data due to privacy concerns and lack of trust. These limitations support generating the synthetic data that may be used for training purpose for providing optimized and robust solutions for increasing industrial efficiency. Mainly, we are contributing to the efficient industrial production of denim jeans in the following perspectives. The generated synthetic images help address the data scarcity problem in defect detection by creating diverse and high-quality training datasets. This optimizes the robustness of defect detection models. Industrial settings often struggle with collecting and labeling a sufficiently large dataset of defective samples. Our approach provides a scalable solution to generate defect variations, which can be used to improve machine learning-based inspection systems. Researchers, manufacturers can integrate these synthetic images into artificial intelligence (AI)-driven quality control systems, leading to improved decision-making and reduced inspection costs.

Technical findings of the underlying work revealed that its better to use 256 × 256 window size instead of 128 × 128 for synthetic image generation with 2,000 epochs. Moreover, DCGAN performed exceptionally well on the broken stitch defect, the skipped stitch defect generation remained on second, while the broken loop defect generation was the most challenging, with more complexity and features in the images. As visually demonstrated in Fig. 10 to Fig. 14 the synthetic images vary in terms of color, size, and orientation of defects such as differently sized broken or skipped stitches and loops appearing at varied angles. These visual variations reflect the diversity introduced by the generator and help in reducing overfitting.

In recent realm, researchers have incorporated synthetic image for providing solutions to optimize the manufacturing or production process. We are also aiming to use these synthetic images for robust denim jeans defect detection in future.

## CONCLUSIONS

The proposed DCGAN architecture demonstrated effective performance in generating various denim jeans defects. The synthetic data accurately captures the features present in the original dataset and can contribute significantly to automatic defect detection systems. While the discriminator network in a GAN model is typically designed to differentiate between original and synthetic samples, the generator in this research has gained sufficient insight into the latent space of the data to deceive the discriminator. The generated images address the challenge of insufficient datasets for broken loops, broken stitches, and skipped stitches in stitched denim jeans pants.

The generator component of the network was designed to introduce diversity into the synthetic image set to avoid overfitting of detection and classification models. The effectiveness of the DCGAN was evaluated using both algorithmic and non-algorithmic methods. Encouraging comments from industry experts and the low FID score indicate the excellent performance of the proposed model.

While this research focused on the most common and basic defects in stitched denim jeans pants, future research could explore other types of denim jeans defects to contribute to a more effective and efficient fully automated defect detection and classification system. Additionally, a wider range of evaluation metrics could be considered in future studies.

## ACKNOWLEDGEMENTS

The authors acknowledge the domain experts for helping in evaluating the generated images.

### Funding

This work was supported and funded by the Deanship of Scientific Research at Imam Mohammad Ibn Saud Islamic University (IMSIU) (grant number IMSIU-DDRSP2503). The funders had no role in study design, data collection and analysis, decision to publish, or preparation of the manuscript.

## Grant Disclosures

The following grant information was disclosed by the authors:
Deanship of Scientific Research at Imam Mohammad Ibn Saud Islamic University (IMSIU): IMSIU-DDRSP2503.

## Competing Interests

The authors declare that they have no competing interests.

## Author Contributions

- Muhammad Naeem conceived and designed the experiments, performed the experiments, analyzed the data, performed the computation work, prepared figures and/or tables, authored or reviewed drafts of the article, and approved the final draft.
- Qaisar Abbas conceived and designed the experiments, authored or reviewed drafts of the article, and approved the final draft.
- Haseeb Ahmad conceived and designed the experiments, performed the experiments, analyzed the data, performed the computation work, prepared figures and/or tables, authored or reviewed drafts of the article, and approved the final draft.
- Muhammad Salman Naeem conceived and designed the experiments, performed the experiments, prepared figures and/or tables, authored or reviewed drafts of the article, and approved the final draft.
- Mutlaq B. Aldajani analyzed the data, authored or reviewed drafts of the article, and approved the final draft.
- Hussain Dawood conceived and designed the experiments, authored or reviewed drafts of the article, and approved the final draft.
- Muhammad Awais Hussain conceived and designed the experiments, performed the experiments, prepared figures and/or tables, authored or reviewed drafts of the article, and approved the final draft.

## Data Availability

The data is available at Zenodo: Muhammad, N. (2025). DenimJeansDefectV2 [Data set]. Zenodo. https://doi.org/10.5281/zenodo.16624901.

The code is available at GitHub and Zenodo.

- https://github.com/MNAInqlabi/DenimJeansDefects.

- MNAInqlabi. (2024). MNAInqlabi/DenimJeansDefects: New_Release_1 (1.0.1). Zenodo. https://doi.org/10.5281/zenodo.14277160.

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
