# Peer review of "DCGAN-based synthetic image generation of denim jeans defects"

_PeerJ Computer Science, doi:10.7717/peerj-cs.3167_

## Round 0.1 · original submission · Major Revisions

Dear Author,
Your paper has been revised. It needs major revisions before being accepted for publication in PEERJ Computer Science. More precisely:

1) You should provide a detailed explanation of the basis and reasons for selecting only these three types of defects as enhancement objects among various types to avoid confusion for readers.

2) In the Proposed DCGAN section, you should provide a detailed explanation of the technical methods and innovations which was proposed or adopted that differ from those existing models.

3) You must avoid repeating the duplicate content in the previous and following sections. Figure 5 and its description (the same as below) should be placed more appropriately in the section on Theoretical Background.

Reviewer 1 ·

Basic reporting

In producing a variety of 758 denim jeans flaws, the suggested DCGAN architecture showed good performance. The synthetic data can make a substantial contribution to automatic flaw identification systems and faithfully captures the features seen in the original 759 dataset. Although the 760 Discriminator network in a GAN model is often made to distinguish between synthetic and original 761 samples, the Generator in this study has learned enough about the data's latent 762 space to trick the discriminator.

Experimental design

The results of the proposed method were found to be satisfactory and reliable.The parameter analysis of the obtained results should be given in more detail. Why have only a limited number of defects been studied despite the large number of existing defect types?

Validity of the findings

The results of the proposed method were found to be satisfactory and reliable.The parameter analysis of the obtained results should be given in more detail.

Additional comments

Literature research should be expanded. Review studies on fabric defect detection should be examined. The deficiencies of the method should be emphasized. Analytical discussions of the results obtained should be included.

Cite this review as

Reviewer 2 ·

Basic reporting

The literature references are not sufficient; defects besides denim jeans should also be discussed.

Experimental design

The types of denim jeans defects should be increased

Validity of the findings

Should be discussed in much deeper detail

Additional comments

1. Since this paper uses DCGAN to enhance three types of defects: Broken Loops, Broken Stitches, and Skipped Stitches, and it is pointed out in the article that “These defects are commonly observed in stitched denim jeans” (line 607). So, what is the significance of choosing them as enhancement objects? In addition, these defective samples are indeed easy to obtain, whether through direct sampling or artificial manufacturing. The author should provide a detailed explanation of the basis and reasons for selecting only these three types of defects as enhancement objects among various types, to avoid confusion for readers.

2. The lines 383-391 describe the composition and working process of the conventional GAN, followed closely by Figure 5, but they appear in the Methodology - Proposed DCGAN. This is not reasonable. The above content has been described in the section on Generative Adversarial Networks in Theoretical Background. Similar situations also appear in many subsequent sections of the Methodology. Please avoid repeating the same content in previous and subsequent chapters. Figure 5 and its description (the same below) should be placed more appropriately in the section on Theoretical Background. In the Proposed DCGAN section, the author should provide a detailed explanation of the technical methods and innovations which was proposed or adopted that differ from those existing models.

3. The line 690 declares “Table 5 presents the FID scores…” but the 693 line shows Table 3 instead. However, Table 5 presents the FID results of generated samples only, while the corresponding data of the original images are missing. Consequently, the comparison between them is not intuitive, and the enhancement effects are not reflected. It is highly suggested that the comparative experiments should be established, and the intuitive quantitative improvements need to be represented to verify the effectiveness of the proposed model.

4. It has claimed that “The Generator component of the network was designed to introduce diversity into the synthetic image set to avoid overfitting of detection and classification models,” from lines 765-766 in the Conclusion, but the diversity of the generated samples has never been mentioned. In addition, the effects of sample enhancement on detection models are not interpreted in the text in like wise. In that way, how did this conclusion come about? To provide evidence, the diversity of the augmented dataset should be proven, and detection outcomes also need to be exhibited.

Cite this review as

---

## Round 0.2 · accepted · Accept

Dear Aour paper has been revised. It has been accepted for publication in PEERJ Computer Science. Thank you for your fine contribution.

Reviewer 1 ·

Basic reporting

The revised article has been improved in terms of content and presentation compared to the first version.

Experimental design

The experimental setup is objective and consistent with the literature.

Validity of the findings

The findings and results obtained are reasonable and scientific. A contribution to the literature has been made.

Additional comments

The developed method is a good example of the use of artificial intelligence methods in fabric defect detection. The findings and discussions are interesting for the textile industry. Original contributions to the relevant field are presented.

Cite this review as